# Phenotypic Variation in *Staphylococcus aureus* during Colonisation Involves Antibiotic-Tolerant Cell Types

**DOI:** 10.3390/antibiotics13090845

**Published:** 2024-09-05

**Authors:** Chloe M. Burford-Gorst, Stephen P. Kidd

**Affiliations:** 1Department of Molecular and Biomedical Sciences, School of Biological Sciences, The University of Adelaide, Adelaide, SA 5005, Australia; chloe.burford-gorst@adelaide.edu.au; 2Research Centre for Infectious Diseases (RCID), The University of Adelaide, Adelaide, SA 5005, Australia

**Keywords:** *Staphylococcus aureus*, coagulase-negative Staphylococcal species, antibiotic tolerance, small colony variants

## Abstract

*Staphylococcus aureus* is a bacterial species that is commonly found colonising healthy individuals but that presents a paradoxical nature: simultaneously, it can migrate within the body and cause a range of diseases. Many of these become chronic by resisting immune responses, antimicrobial treatment, and medical intervention. In part, this ability to persist can be attributed to the adoption of multiple cell types within a single cellular population. These dynamics in the *S. aureus* cell population could be the result of its interplay with host cells or other co-colonising bacteria—often coagulase-negative Staphylococcal (CoNS) species. Further understanding of the unique traits of *S. aureus* alternative cell types, the drivers for their selection or formation during disease, as well as their presence even during non-pathological colonisation could advance the development of diagnostic tools and drugs tailored to target specific cells that are eventually responsible for chronic infections.

## 1. Introduction

*Staphylococcus aureus* is one of the most notorious bacterial species that are globally responsible for significant infection-related morbidity and mortality, with a fatality burden that consistently exceeds over one million deaths annually [1]. This reality is an unfortunate consequence of *S. aureus’s* ability to cause acute and chronic infections presented across a broad spectrum of diseases, from common skin and soft tissue infections to the more severe bacteraemia, pneumonia, osteomyelitis (OM), complications with cystic fibrosis (CF), and endocarditis, which can all lead to sepsis or death [2]. However, *S. aureus’s* success as a microorganism is not just a result of its ability to cause disease; instead, it can be largely attributed to *S. aureus’s* remarkable ability to adapt and colonise a range of anatomical niches within a human host [3]. *S. aureus* has been found to intermittently colonise approximately 30% of the healthy human population, predominantly in the mucous membrane, anterior nares, and the skin, and possibly also in the gut [4]. Although these colonising strains of *S. aureus* are asymptomatic, their opportunistic nature causes the bacteria to concurrently possess the ability to invade other sites in the body and cause disease to its human host, often when the immune system is compromised [5]. Colonising *S. aureus* contributes to the infection rates caused by community-acquired and nosocomial infections, as well as causes endogenous infections where the bacteria can transit through its host [6]. *S. aureus’s* prevalence and ability to become a pathogen means it is essential to implement precautions and consider the risk factors of *S. aureus* infections in individuals with impaired immune systems, such as those with co-morbidities and chronic pathologies (patients with diabetes, heart disease, or chronic obstructive pulmonary disease (COPD)), those on dialysis, and those undergoing surgery [7,8,9,10]. In addition, it is not only necessary to understand *S. aureus’s* molecular and genetic factors that drive its virulence during the specific diseases it causes, but it is also important to strengthen our understanding of the factors that enable its colonisation, which is a major risk factor for these patients.

Further to its transit into various human tissues, *S. aureus* harbours numerous genetic loci that enable its resistance to antibiotic treatment. The development and transfer of these genes increases the complexity in treatment regimes. Combined with its dynamic metabolic range, specific virulence factors and the possession of these resistance genes underpin much of the chronic nature of *S. aureus* infections. However, it has become apparent in recent years that the failure of medical intervention in *S. aureus* infectious diseases, and thereby the basis for their relapse, is not limited to the possession of a specific set of virulence or antibiotic resistance genes but includes a broader shift in the bacterial cell types that permits a small number of quasi-dormant *S. aureus* cells to hide from the immune system and tolerate antimicrobial treatments. This review will present the current understanding of the development of these cell types during colonisation and extend the knowledge on antibiotic resistance to the phenotypic variation as a core attribute of subsequent chronic and relapsing infections.

## 2. Colonisation of Humans by *Staphylococcus aureus*

*S. aureus* has been found to have a distinct preference for specific areas during human colonisation, primarily the anterior nares, gastrointestinal tract, skin (most commonly the axillae and groin), throat, and vagina [11]. The anterior nares have long been established as the major reservoir for colonising *S. aureus*, with approximately 20–30% of the healthy population being permanent asymptomatic carriers and approximately 60–75% being intermittent carriers [4,12]. In contrast, healthy human skin is thought to offer unfavourable conditions for the establishment of permanent *S. aureus* skin colonisation. The basis for this remains unknown; however, it is speculated that the large surface area of the skin contributes to a lower density of colonising microbes that are constantly competing for regularly fluctuating nutrient availability and defects within the skin barrier [13]. It is also speculated that *S. aureus* is more likely to only temporarily colonise particular areas of the body, such as the skin, as a result of the bacteria hiding within or permanently colonising other niches of a single person. *S. aureus’s* successful colonisation of different sites around the human body is attributed to the specific molecular and genetic adaptations that allow it to adhere to its human host and evade therapeutic and host-generated stresses.

### 2.1. S. aureus and Mechanisms for Colonisation of Human Tissue

During nasal and skin colonisation, *S. aureus* has been found to highly express genes encoding for microbial surface components recognising adhesive matrix molecules (MSCRAMMs) (*sdrC*, *sdrD*, *sasG*, *isdA*, *atlA*, *clfB*, *eap*, *fndA*), cell surface dynamic and remodelling enzymes (*atlA*, *sceD*, *oatA*), wall teichoic acid (WTA) biosynthesis (*tarK*, *tagO*), and immune-modulatory factors (*spa*, *chp*, *sap*, *sak*) [14,15,16]. In contrast, *S. aureus*’s quorum-sensing accessory gene regulator (*agr*) system, which is responsible for the induction of protease and cytolytic toxin production and the downregulation of surface proteins, has been found to have reduced expression in colonising strains [16,17,18]. As a result, genes encoding for several toxins, including haemolysins (*hla*, *hld*, *hlg*), phenol-soluble modulins (*psm*), and bi-component leukotoxin homologue (*blhB*), are downregulated [17,18,19] (see Figure 1).

Unlike when colonising the nose, *S. aureus* within the skin microbiome must overcome the physical permeable barrier and chemical antimicrobial challenges that make up the human skin. An example of one of these stressors is the excretion of antibacterial fatty acids (AFAs) produced by sebaceous glands [20]. To overcome this, *S. aureus* over-expresses genes encoding for WTA and the iron-regulated surface determinant A (IsdA), which have both been shown to protect the bacteria from AFAs [20,21]. IsdA has also been shown to protect *S. aureus* from antimicrobial peptides (AMPs) in nasal secretions by binding to lactoferrin and subsequently inhibiting its proteolytic activity [22]. In addition, colonising *S. aureus*’s ability to express several adhesion proteins highlights the multi-functional nature of its interactions with its host. The abundance of different adhesive proteins presents a challenge when developing prophylactic treatments (such as vaccines) for decolonisation, as the selection of one or two proteins (e.g., ClfB or IsdA) could easily result in the bacteria employing other adhesins for colonisation [23]. A previous study analysing the gene expression of *S. aureus* during colonisation over 10 days showed variations in the expression of adhesive proteins at different stages of colonisation. Genes involved with WTA biosynthesis were found to be important during the early stages of colonisation, suggesting its importance for the initial bacteria–host interactions that bind to the scavenger receptor class F (SCRE-1) expressed by epithelial cells [24], whereas *clfB* and *isdA* were upregulated in the later stages, indicating their importance for maintaining attachment to the host tissue [18,19,25] by binding to iron-regulated surface determinant A (IsdA) [26], as well as for interactions with cytrokeratin and keratinised nasal cells (such as cytokeratin 10) [27] and the cell envelope protein loricrin [23].

This temporal regulation of genes during colonisation permits *S. aureus* to accumulate over time, allowing it to potentially regulate specific genes (such as via quorum-sensing and virulence pathways using Agr) and thereby switch to a pathogenic state when the host’s immune defences are compromised. This transit from colonisation to the infection site can be caused by aspiration, surgery, shaving, or catheter insertion [11,12,28,29]. These infections can therefore be caused by the bacteria already existing within the host (initially as commensals) and are referred to as autoinfections or endogenous infections. As such, individuals who are either persistent or intermittent carriers of *S. aureus* are at a higher risk of disease when their immune systems become compromised [29]. Additionally, it has been shown that polymorphisms within the host genome, such as interleukin-4 (IL-4), C-reactive protein (CRP), complement factor H (CFH) [30], defensins [31], mannose-binding lectin genes [32], and Toll-like receptors (TLRs) [32,33], frequently correlate to the colonisation of *S. aureus*, but there are only a small number of studies on the relation between host genes and *S. aureus* colonisation, leaving much nuance regarding the impact of these host genetic polymorphisms and the heritability of *S. aureus* colonisation [33].

### 2.2. Interactions between Colonising Staphylococcal Species

The human body harbours a rich array of densely populated microorganisms that interact and compete with one another through a range of mechanisms in order to uphold homeostasis and successfully colonise [34]. Different anatomical niches within the human host, including the nose, skin, and gastrointestinal tract, harbour unique microbiomes that differ in their phylogenetic and functional gene compositions [5]. Research efforts focusing on the nasal and skin microbiome have found that both environments are predominately populated by the *Corynebacterium*, *Propionibacterium*, *Streptococcus*, and *Staphylococcus* genera of bacteria [35]. From the colonising *Staphylococcus* species, the most commonly identified members include not only *S. aureus* but also the coagulase-negative Staphylococci (CoNS) species *S. epidermidis*, *S. lugdunensis*, *S. capitis*, *S. warneri*, *S. haemolyticus*, and *S. hominis* [36]. These CoNS species that are in direct competition with *S. aureus* have developed unique survival tactics to either inhibit or outright kill *S. aureus* without invading and causing infection to the host directly (Figure 2) [37]. These mechanisms include the biosynthesis of lantibiotics and bacteriocins. Recent studies have focussed on these bacterial compounds and their role in allowing for their dominance of local environments, such as the anterior nares and skin, specifically using these compounds also to target MRSA (methicillin-resistant *S. aureus*) and MDR (multi-drug-resistant) *S. aureus*.

#### 2.2.1. *Staphylococcus epidermidis*

Some strains of *S. epidermidis* have been found to synthesize and release extracellular serine protease (Esp). Esp has been found to degrade various proteins that are essential for adhesion to host surfaces and cohesive attachment to other bacteria for biofilm formation during colonisation [38]. The specific bacterial surface proteins degraded by Esp during nasal colonisation shown in in vitro and in vivo studies include adhesion molecules (SdrD, Emp, Eap, IsdA), cell surface dynamic and remodelling enzymes (Atl, SceD), immune-modulatory factors (Spa, Sbi), fibronectin-binding proteins (FnBPA, Efb), and β-haemolysin (Hlb). Esp was also found to degrade host receptor proteins, such as fibronectin, fibrinogen, and vitronectin, which are also important for the colonisation of *S. aureus* [39]. Through this action, Esp therefore acts indirectly for *S. epidermidis* to out-compete *S. aureus* within these host sites. Similar to interactions between Esp and *S. aureus* in the anterior nares, *S. epidermidis* has also evolved strategies to inhibit or kill other competing bacteria during skin colonisation. In particular, *S. epidermidis* expresses molecules that target colonising *S. aureus*. These include PSMs (PSM-γ and PSM-δ) that cause membrane leakage, cognate autoinducing peptides (AIPs) that disrupt quorum sensing by inhibiting the Agr system [40,41], and bacteriocins (Epidermin, Pep5, and Epilancin K7) that disrupt the barrier of the microbial cytoplasmic membrane [5].

#### 2.2.2. *Staphylococcus lugdunensis*

*S. lugdunensis* is a prominent example of a CoNS species that generates a potent antimicrobial peptide and does not out-compete its bacterial neighbours. Specifically, it synthesises lugdunin, which has a bactericidal effect on colonising *S. aureus* [42]. Lugdunin is a lantibiotic that is synthesised by a Non-Ribosomal Peptide Synthase (NRPS) operon, which consists of four NRPS genes (*lugA*, *B*, *C*, D) [42]. In the anterior nares, lugdunin inhibits target microbes by dissipating their membrane potential, and it is theorised to achieve this using a protonophore-like mechanism [43]. Additionally, lugdunin is also capable of stimulating host skin cells to produce peptides that have antimicrobial properties and, when in synergy with lugdunin, can eliminate *S. lugdunensis*’s neighbouring and susceptible bacteria [44].

#### 2.2.3. *Staphylococcus warneri*

The interactions between *S. warneri* and *S. aureus* have not been greatly researched. However, a recent study examined the mechanisms used by various colonising CoNS species to interfere with the quorum sensing of *S. aureus* through intra- or interspecies cross-talk. It was found that *S. warneri* secretes AIPs, specifically AIP-II, which has a potent inhibitory effect on the MRSA Agr system. *S. warneri* AIP-I still inhibited Agr but had a much weaker interaction with the *S. aureus* Agr system [45]. Coinfection experiments with the two bacteria also showed that *S. warneri* was capable of preventing transepithelial water loss and ultimately preventing barrier erythema and scaling, a strong indicator that *S. warneri* and its AIPs repress the Agr system of *S. aureus* and, consequently, the production of its virulence factors during invasion, and possibly also its colonisation of the skin. AIP-II was ultimately determined to be an effective inhibitor of MRSA agr-IV, with the capability of reducing the dermo-necrotic lesion size during skin infections [46]. It is important to take into account that there are no other studies that have investigated the interactions between *S. aureus* and *S. warneri*, or how *S. warneri* may play an important role in preventing or controlling *S. aureus* growth during colonisation and/or invasion. Thus, there is clearly still much nuance that remains unknown around the role that *S. warneri* plays as part of the CoNS commensal species and its subsequent antimicrobial effects on potentially pathogenic bacteria such as *S. aureus*.

#### 2.2.4. *Staphylococcus aureus* Interactions with CoNS Species

In addition to the previously detailed CoNS species, *S. aureus* has also been found to have direct and dynamic interactions with other CoNS species that can inhibit or kill colonising *S. aureus*, such as *S. capitis*, *S. chromogenes*, *S. pseudintermedius*, and *S. epidermidis*, which have all been found to synthesise the purine analog 6-thioguanine (6-TG), a molecule found to inhibit *agr* quorum sensing, resulting in a reduction in the virulence and growth of *S. aureus* [47]. Furthermore, *S. hominis* [48], *S. caprae* [49], and *S. simulans* [50] have also been shown to produce novel, short autoinducing peptides that interfere with and ultimately block *S. aureus* quorum sensing, resulting in a reduction in bacterial growth [45,51].

*S. capitis* has also been found to produce a bacteriocinbiotic known as gallidermin, which can bind to the cell wall precursor lipid II of *S. aureus* and inhibit cell wall synthesis [52,53]. Alternatively, it has also been shown to inhibit WTA synthesis (which, as previously discussed, is important in the early stages of colonisation) [54]. *S. hominis* also secretes a bacteriocin known as micrococcin P1, which has been demonstrated to not only reduce *S. aureus* growth during infection but also to accelerate *S. aureus*-infected wound healing [55,56].

## 3. Changes in *S. aureus* Lifestyle during Colonisation

The variation in cell types within a single population of *S. aureus* cells during its different infections (such as OM and CF) have been a focus in numerous studies specifically directed at understanding chronic and relapsing infections. However, it is hypothesised that the presence of persistent cell types such as small colony variants (SCVs) and those that are predisposed to forming biofilms are not limited to *S. aureus* during a pathology but also exist within a population of colonising *S. aureus*. The ability to switch to a quasi-dormant cell type (or for such a cell type to exist as a subpopulation) provides multiple advantages in the overall survival of the bacteria, mainly via a slower metabolic rate. This consequently allows *S. aureus* to adapt to changes in different environmental stresses within the host, such as the presence of antibiotics, pH levels, competition with other colonising bacteria, and nutrient starvation stresses [57]. Unlike the growth rate of typical *S. aureus*, these quasi-dormant subpopulations are defined by their slow metabolic rate but prolonged survival under physical and chemical stresses, as well as their ability to live within (intra- and extracellular) host tissue, often without an immune response. Currently, the incidence of these subpopulations is well described for specific *S. aureus* infections; however, the presence of these cell types of *S. aureus* during colonisation is not well understood.

### 3.1. S. aureus and Biofilm during Colonisation

*S. aureus’s* ability to form complex biofilms is a contributing factor to the bacteria’s pathogenesis. Biofilms frequently form on the artificial surfaces of biomedical devices (such as medical implants and catheters) but have also been found to form on host tissues and during colonisation [58,59]. These biofilms are characterised by static, multiple layers of spatially grown heterogeneous single cells and microcolonies held together by a matrix referred to as an extracellular polymer substance (EPS) [60,61,62]. This matrix variously comprises proteins, extracellular DNA (eDNA), polysaccharide intercellular adhesin (PIA), and amyloid fibrils [60,62,63].

Colonisation, for example, of the skin, by *S. aureus* is initiated through adhesion by its surface-anchored proteins, such as SasG, to human cell types, particularly corneocytes. SasG (and other surface proteins that act as adhesins) also function in adhesion to nasal epithelial cells and are known to be functional in biofilm formation. Indeed, there are a large number of *S. aureus* adhesins that bind to human extracellular matrix (hECM) proteins and act as the beginning for biofilm formation and colonisation. However, *S. aureus* also has the ability to form planktonic aggregates, which are similar to biofilms but are distinct in composition and activity. Although biofilms are well characterised during invasion and are linked to persistent infections, their role in colonisation is not definitive. Currently, there is some evidence that biofilms are important in different sites for colonisation, but this seems to be transient, and there are contradictory reports that *S. aureus* forms biofilms during colonisation [5,37,64]. Dense bacterial populations (such as those in biofilms) have been shown to activate the Agr system (which, as previously discussed, is downregulated during colonisation), which allows for the expression of toxins and other proteins that favour an invasive phenotype [18,19]. Specific bacterial species are also commonly found to co-colonise human hosts with *S. aureus*, and these can secrete specific extracellular molecules that inhibit biofilm formation, such as the previously discussed ESP expressed by *S. epidermidis* [39,65,66,67].

Some studies have shown that *S. aureus* is dispersed and not consistent with a biofilm during nasal colonisation. Finally, because nasal-colonising *S. aureus* can be eradicated using mupirocin, it is presumed that biofilms do not form in the nasal cavity, as it would be likely that they would persist even after antimicrobial intervention [68]. However, there have been studies that have produced results showing that during colonisation, there are direct interactions between *S. aureus* and other bacteria that favour single-species *S. aureus* biofilms and promote polymicrobial biofilm formation, such as the co-colonisation of *S. aureus* and *Propionibacterium* spp. as a result of *Propionibacterium*-produced coproporphyrin III (CIII), which has been shown to induce biofilms and the coaggregation of *S. aureus* [69]. Due to the low number of bacteria, such as *S. aureus*, that populate the skin and nose, it is heavily suggested that *S. aureus* appears predominately as small clusters or dispersed cells [19].

### 3.2. S. aureus Small Colony Variants

SCVs are stochastically generated as a subpopulation within a normal population of actively growing cells. SCV cells exist within this population with a slow growth rate and an atypical colony morphology and have antibiotic tolerance as part of a population of antibiotic-susceptible cells [70]. Their phenotypic characteristics have been identified for over 100 years from clinical samples, with colonies observed to be slow-growing, non-pigmented, and approximately 1/10th the size of typical colonies (Figure 3). They also have reduced coagulase activity, reduced hemolysin production, increased antibiotic tolerance, and decreased membrane potential [71,72]. As a result of these characteristics, SCVs are often misdiagnosed as not *S. aureus*. The persistent phenotypic variation in SCVs has also been well described, such as their presence during *S. aureus* infections from soft tissue infections, bacteraemia, CF, OM, brain abscess, and device-related infections [71,73]. They have also been directly linked to unsuccessful treatment with long-term antibiotics [74], having been shown to tolerate a range of antibiotics, including fluoroquinolones, trimethoprim–sulfamethoxazole (SXT), fusidic acid, and aminoglycosides [75]. Although SCVs are well studied, they remain a clinical challenge to overcome when treating chronic *S. aureus* patients, or even in the development of a vaccine to decolonise *S. aureus* in healthy populations. SCV development is impacted, and formation is enhanced, by a variety of alternative metabolic pathways. These changes in metabolism have been related to various auxotrophisms, reduced ATP production, downregulated membrane potential, reduced CO_2_ biosynthesis, changes in the function of global regulatory proteins, reduced cell wall biosynthesis, and reduced fatty acid biosynthesis (Figure 4) [71,76,77,78].

Further complicating studies and clinical practice, there are different versions of SCVs. Electron transport-defective SCVs have a decreased biosynthesis of two essential components involved in the electron transport chain: menadione, essential in the biosynthesis of menaquinone caused by a mutation in *menD* and then in *hemB* (haemin), a porphyrin used in cytochrome biosynthesis [79,80]. Other mutations that affect electron transport include a mutation in *ctaA*, inhibiting haem A biosynthesis, which is also important for cytochrome biosynthesis [57,81]. Whilst SCVs with mutations in menadione and haemin have been isolated in patients suffering from infection, especially whilst undergoing treatment with aminoglycosides [74], the phenotypic variation displayed by these SCVs has been shown to be reversible through supplementation with the defective element, thereby supporting the clinical relevance of defects in the biosynthetic pathway of menadione and haemin [71,78,82]. Studies into *hemB* mutants in *S. aureus* SCVs have also shown that these have a reduction in α-toxin production [83], as well as the increased expression of *clfA* and fibronectin-binding protein (*fnb*) genes [84,85]. The increase in *clfA* and *fnb* allows for the mutant SCVs to have a greater ability to bind to the fibrinogen and the fibronectin of the host cells [84,85,86]. The *ctaA* mutant found in some SCVs has been shown to produce phenotypic variation within the colonies not only by a reduction in the colony size, but also by a decrease in the toxin production of α-toxin and toxic-shock-syndrome toxin 1 (TSST 1), a decrease in the expression of RNAIII, and an increase in resistance to aminoglycoside antibiotics [81].

Thymidine biosynthesis-defective SCVs are primarily related to patients with CF and patients undergoing long-term treatment with SXT [87]. The production of these SCVs is stimulated by SXT, as it interferes with the synthesis of tetrahydrofolic acid, an enzyme essential in the synthesis of thymidylate synthase encoded by the *thyA* gene. Unlike typical SCVs, thymidine-dependent SCVs can also exhibit two different phenotypical growth characteristics: the typical pinpoint colonies and colonies with an elevated, pigmented centre surrounded by a translucent edge on Columbia blood agar. These are referred to as “fried egg” colonies [88]. These phenotypes are also reversible by supplementation with thymidine [89].

In addition to auxotrophic SCVs, the *S. aureus* MazEF toxin–antitoxin system is also suspected to be involved in the generation of SCVs. This toxin–antitoxin system is transcribed as an operon where MazF (toxin) is readily bound to MazE (antitoxin), thereby neutralising the self-harming toxicity of MazF [57]. The antitoxin MazE is an RNase that interacts with *hla*, *spa*, and *sigB* mRNAs whilst avoiding *sarA*, *recA*, and *gyrB* mRNAs [90]. Increased production of MazF (beyond the capacity of MazE) gives rise to the typical SCV phenotype with a decreased growth rate and pigmentation with an increased survivability under various stresses [91]. Similarly, with an increase in the ATP-dependent protease system, ClpCP targets and reduces the MazE levels, allowing for an increase in MazF [70,92].

SCVs have increased expression of downregulators of the accessory gene regulator (*agr*) pathway, such as SrrAB, CodY, SigB, PSMs, and ArIRS. Meanwhile, positive regulators of Agr exhibit either reduced expression (SarA) or their expression is inhibited (MgrA) [75,93,94,95,96,97,98]. These changes in the regulation of the Agr operon ultimately result in a reduction in RNAIII production and therefore the inhibition of toxin expression, such as *hla* and TSST-1 [99,100]. However, there are exceptions to this pattern, with the SarA expression being downregulated in thymidine-dependant SCVs whilst being upregulated in *hemB* and *menD* mutants [101].

There are other genetic changes found to be associated with different SCVs, including mutations in *relA*, which encodes for RelA hydrolase. A reduction in the activity of RelA hydrolase results in the accumulation of the guanosine 3′,5′-bis(diphosphate) (ppGpp), an intracellular signalling molecule that causes the permanent activation of the stringent response [102]. Other enzymes that increase the synthesis of ppGpp are also dysregulated by Rsh, RelP, and RelQ [94]. The permanent activation of the stringent response limits protein synthesis, thereby reducing cellular growth and producing small colonies, and it is directly linked to linezolid resistance [102,103].

### 3.3. Selection for Small Colony Variants within the Polymicrobial Microbiome

An important developing area of understanding is the nature of polymicrobial niches that influence the phenotypic variation within *S. aureus* populations, such as those observed in patients with CF [104,105]. These studies have observed that signal molecules released by *Pseudomonas aeruginosa*, such as 4-hydroxyl-2-heptyl quinoline-N-oxide (HQNO), hydrogen cyanide, and pyocyanin, can protect *S. aureus* against aminoglycosides [106]. *P. aeruginosa* releases these molecules as secondary metabolites and these act as respiratory inhibitors of the growth of *S. aureus* by inhibiting *S. aureus* electron transport [104,106]. This does not impact SCVs (already not using the electron transport chain) and creates favourable conditions for SCV cells to continue to survive and thereby dominate within a population of *S. aureus* cells.

Within an anatomical niche, the constant battle for survival between colonising bacterial species creates different stresses that can be detrimental to the success of actively growing *S. aureus.* As previously established in this review, CoNS such as *S. epidermidis*, *S. lugdunensis*, and *S. warneri* can produce exoproducts, such as bacteriocins, proteases, AIPs, and PSMs, that can inhibit or kill *S. aureus*.

In addition to the stresses from CoNS species, *Streptococcus pneumoniae* is able to kill *S. aureus* due to its production of hydrogen peroxide [107,108,109]. Hydrogen peroxide causes irreversible damage to specific protein groups by oxidising iron groups or product OH^-^, resulting in DNA damage. Much like the selective pressure from HQNO from *P. aeruginosa*, hydrogen peroxide has also been shown to select for aminoglycoside-resistant SCVs that have mutations predominantly in *menD* and less so in *hemB* [110]. It has also been shown that hydrogen peroxide, when used as a method of decolonisation, is unsuccessful at eradicating *S. aureus* in the nasal cavities of neonatal rats [111].

*Cornyebacterium* spp., like *S. aureus*, has also been well established as a commensal within the nasal cavity [35]. *Cornyebacterium* spp. (such as *C. pseudodiphtheriticum*) have been shown to inhibit the *S. aureus* Agr system suspected to be the result of the expression of inhibitory AIPs or PSMs [112]. As has already been well established, *S. aureus* downregulates *agr* in order to favour colonisation factors, biofilms, and SCVs [113,114], and the inhibition of the agr system from other bacterial species can be problematic to individuals who are immunocompromised. Alternatively, *Cornyebacterium* spp. can directly kill *S. aureus* and therefore potentially drive it to turn off its virulence gene expression in favour of a more persistent lifestyle (SCV) and to escape being killed [112].

## 4. Antibiotic Responses of *S. aureus* Lifestyles

### 4.1. Antibiotic Resistance by S. aureus Strains

A wide range of *S. aureus* species have evolved over time to develop specific mechanisms that allow them to survive and continue growing in the presence of different antibiotics [104]. These antibiotic-resistant strains of *S. aureus* have become highly prevalent, especially in hospital-acquired infections. The most significant strains with an impact on both hospital- and community-acquired bacterial infections are the MRSA strains [115]. Methicillin, like penicillin, is a β-lactam class of antibiotics, and resistance occurs through the acquisition of the *mecA* gene located on the Staphylococcal Chromosomal Cassette (SCC*mec*). This SCC element is a large fragment of DNA that is categorized as a mobile genetic element (MGE) and often encodes antibiotic resistance and/or virulence determinants that can be passed on between *S. aureus* populations through horizontal gene transfer [115,116]. As well as MRSA, another common antibiotic-resistant *S. aureus* population is vancomycin-resistant *S. aureus* (VRSA). Vancomycin is a glycopeptide class of antibiotic for which resistance arises when the developing mutations in the *vanA* operon, which consists of the *vanRSHAXYZ* genes, cause modifications to the cell wall that prevent vancomycin from binding efficiently [117,118]. *S. aureus* has also been found to develop resistance to other antibiotic classes, including but not limited to aminoglycosides, tetracycline, oxazolidinone, macrolides, fluoroquinolones, rifampicin, and lipopeptides [119,120,121], summarised in Table 1. An equal concern is the growing population of the different multi-drug-resistant (MDR) strains of *S. aureus* that have complete resistance to three or more antibiotic classes, for example, *S. aureus* strains that are resistant to oxacillin (β-lactam), ciprofloxacin (fluoroquinolones), linezolid (oxazolidinone), and erythromycin (macrolide) [122].

### 4.2. Antibiotic Tolerance: Indirect and Direct Responses by S. aureus to Antibiotics

In contrast to antibiotic resistance, there is the ability of a bacterial strain to temporarily withstand or slow down the lethal consequences of high doses of antibiotics, measured by the decrease in antibiotic killing and resulting in continued bacterial survival [123]. These tolerant bacterial strains are typically genetically susceptible to antibiotic killing [124], unlike antibiotic resistance strains that have mutations in specific genes that allow them to resist antibiotic treatment by a broad shift in the bacterial phenotype. Tolerance can arise from a stochastic change in metabolic functions, causing a reduction in antibiotic uptake and preventing the killing of various antibiotic classes [125]. This phenotype is thought to present itself as either tolerance by “slow growth” (occurring in a steady state) or “by lag” (an extended lag phase of growth, which is a transient state as a response to stress conditions) [126]. An example of an *S. aureus* phenotype that exhibits antibiotic tolerance is the SCV phenotype, which, as previously mentioned, exhibits mutations that favour slow growth and changes in the metabolic pathways that allow for antibiotic tolerance [127]. The antibiotic targets are almost always the molecular mechanism during growth, and in a slow or no-growth state, these targets are not functional. It is currently understood that antibiotic tolerance is a phenomenon synonymous with persister cell phenotypes; however, there is no known identifying genetic marker that can be directly linked to these cells’ ability to tolerate antibiotics [128]. Thus, due to the difficulty in isolating persister cells, and SCVs, from clinical samples, these subpopulations that are tolerant to antibiotic treatments can exist within a population of cells that display antibiotic susceptibility, and can persist and ultimately result in chronic or relapsing infections [129].

### 4.3. Heteroresistance and S. aureus

Defining antibiotic resistance has been a strong focus in the research for the past decades, focusing on mutations and the acquisition of various genetic changes that differ from the typical to the genes in typical susceptible phenotypes [130]. However, there are exceptions to the correlation between the presence of resistant genes and the observed resistant phenotypes. An example of this exception is the phenomenon of heteroresistance, where subpopulations present within a heterogeneous cell population, possessing cells with elevated levels of resistance to antibiotics compared to those of the dominant cell population [131]. Unlike antibiotic-tolerant bacterial strains, heteroresistant populations proliferate under antibiotic stress and can arise either intrinsically (without pre-exposure to antibiotics) or via acquisition (from initial exposure to antibiotics) [132]. This ultimately gives rise to subpopulations of persistent cell types, such as those that are predisposed to form biofilms or even SCVs [133]. The low population of these heteroresistant strains within a single isolate makes it increasingly difficult to reliably determine an accurate susceptibility profile of the bacterial strain. Currently, the methods to determine the presence of heteroresistant strains include population analysis profiling (PAP), Etest, disc diffusion, and agar dilution [134,135]. Despite these established assays, heteroresistant strains remain easy to miss and are suggested to play an important role in prolonged infections and mortality rates [136]. This has become more frequently observed in vancomycin-intermediate *S. aureus* (heteroresistant VISA (hVISA), which includes intermediate strains but with subpopulations that display resistance) within hospital settings [117].

The current understanding is that heteroresistance results from a few different mechanisms can be categorised as either stable (where resistance persists after the removal of the antibiotic stress) or unstable (where the population reverts back to an antibiotic susceptible phenotype after the antibiotic selection pressure has been removed) [135] (Figure 5). Stable heteroresistance is frequently observed to be a result of mutations that are typically genetically stable, such as single-nucleotide polymorphisms (SNPs), frameshifts, insertions, and deletions [135]. These mutations generally have a marginal impact on the fitness cost, allowing the subpopulation to remain resistant to specific antibiotics even once the stress has been removed. These stable phenotypes have been observed in both Gram-positive bacteria such as *S. aureus* (e.g., hVISA) and Gram-negative bacteria such as *Acinetobacter baumannii* [134]. Alternatively, unstable heteroresistance is thought to be a result of spontaneous, genetic, tandem amplification, where an increase in resistance genes present within the bacterial genome gives rise to increased resistance [137]. This unstable phenotype has been almost exclusively observed in Gram-negative bacterial species, such as *Escherichia coli* and *Haemophilus influenzae*, due to the spontaneous homologous recombination between repeated sequences in sister chromatids during replications [138]. Thus, amplified gene regions increase resistance corresponding to an increase in the gene dosage (such as genes encoding for antibiotic resistance, efflux pumps, or antibiotic-modifying/-degrading enzymes) [139]. However, due to the high fitness cost of amplification, the gene is de-amplified when the stress is removed [140].

A recent study looked at the genetic changes in heteroresistant *S. aureus* isolates compared to their parental strains. The prevalence of heteroresistance was studied in 40 parental clinical *S. aureus* isolates when grown in the presence of daptomycin, gentamycin, linezolid, oxacillin, teicoplanin, and vancomycin [141]. Amongst the 40 isolates, 52.5% exhibited heteroresistance to at least one of the antibiotics tested, 17.5% exhibited heteroresistance towards two, and 15% exhibited heteroresistance towards three, with only 15% of isolates exhibiting no heteroresistance to any of the antibiotics. Genetic mutations in each heteroresistant strain isolated were dependent on which antibiotic to which it displayed resistance. Mutations in known antibiotic-resistant genes were most commonly observed as a result of SNPs, frameshifts, insertions, or deletions typical to those seen in stable heteroresistance; however, mutations in genes associated with the heme and menaquinone pathways, transfer RNA (tRNA) modification, tRNA aminoacylation, and the ribosomes were also observed [141]. This study, although small, did illustrate the prevalence of heteroresistance in *S. aureus*, and that the regular mutations within the bacterial chromosome in different core genes are the main mechanisms by which heteroresistance is conferred. They also highlighted that the reason that Gram-positive bacteria, such as *S. aureus*, do not present with random gene amplification is due to the lack of resistance genes and repeat sequences present within the bacterial genome [141].

### 4.4. Clinical Significance of S. aureus Small Colony Variants and Colonisation

There are different and specific contexts to the significance of the variations in *S. aureus* cell types. *S. aureus* is one of the most common causative pathogens of infections worldwide, responsible for a vast array of nosocomial and community infections [142]. Chronic and relapsing infections have been linked to the presence of SCVs and the inability of antibiotics and the immune system to completely remove these cells. However, healthy individuals colonised with *S. aureus* have been shown to have an increased risk of invasive and non-invasive infections [68,143,144,145], with over 80% of *S. aureus* infectious strains found to be genetically similar and endogenous to those isolated from the nares of colonised individuals [6,146,147,148].

*S. aureus* is the most isolated pathogen from diabetic foot infections (DFIs), commonly resulting in persistent OM infections or DFI-OM [149]. It has recently been shown that ~39% (n = 109/276) of DFI patients were nasal carriers of *S. aureus*, whilst ~36% (n = 101/276) harboured *S. aureus* in both their nares and sites of infection. Of these patients, ~65% of *S. aureus* strains were genetically identical in both locations [150]. This has also been seen in previous studies looking at the relationship between nasal and diabetic foot ulcer (DFU) *S. aureus* colonies [104,147,151,152,153]. SCVs have also been recovered from DFI-OM patients [74]; however, the role SCVs have in persistent and relapsing infection is still not well studied.

In contrast to the unknown role of SCVs in DFI-OM by *S. aureus*, SCVs have been extensively studied in CF patients, of which approximately two-thirds are infected with *S. aureus* SCVs in conjunction with *P. aeruginosa* [154,155]. As previously mentioned, coinfection with *P. aeruginosa* and long-term aminoglycoside or SXT treatments creates environmental stresses within the host that select for the SCV phenotype within a population of *S. aureus* [104,156,157,158,159]. SCVs have been directly linked to poor clinical outcomes in CF patients, as they contribute directly to the chronic inflammation of the lungs [104,160]. Although the involvement of SCVs in CF patients has been well researched, the role that colonisation has in contributing to CF infections is not well characterised. This also holds true for device-related (such as prosthetic joints) infections and skin and soft tissue infections, for which the role of SCVs is well researched but the involvement of colonisation in persistence and relapse is not [161].

### 4.5. Diagnosis and S. aureus Small Colony Variants

A range of rapid non-molecular and molecular diagnostic methods are available to identify and diagnose *S. aureus* directly from clinical samples. There are selective and differential media that have been developed to either isolate or select for *S. aureus* by specifically taking advantage of the unique *S. aureus* enterotoxin and enzymes produced by the bacteria. Blood agar is commonly used as a differential media to identify haemolytic characteristics of bacteria that are also typical of *S. aureus*. However, the use of the selective media: *Staphylococcal* medium 110, Brilliance Blue, and Mannitol Salt Agar (MSA), allow for the isolated growth of *S. aureus* with minimal to no growth of *E. coli* over a 16–24 h incubation. Positive colonies can undergo further identification with a PBP2a latex agglutination assay, polymerase chain reaction (PCR) of the *mecA* gene, and susceptibility testing to determine antibiotic resistance [162]. More modern diagnostic techniques include culturing *S. aureus* on the selective medium chromogenic *S. aureus* ID agar, where *S. aureus* strains such as methicillin-resistant *S. aureus* (MRSA) can be detected based on the colour of the colonies present. Finally, *S. aureus* can be identified within 1–4 h through PCR alone from a collected nasal swab [162,163].

However, none of these methods take into account the cells within an *S. aureus* infection, which possess a slow growth rate or very low to no expression of the usual traits, as seen in SCVs. These cells can therefore remain undetected, making standardised testing methods difficult, and SCV-based infections or infections, including SCV cells as part of the infection, are often missed during the clinical identification of *S. aureus* [164]. These atypical *S. aureus* phenotypes are identified as non-haemolytic, non-pigmented, pinpoint colonies due to deficiencies or reductions in biochemical reactions [165]. Figure 3 provides a comparison of typical golden *S. aureus* growth, atypical non-pigmented active cell growth, and SCVs on tryptone soy agar (TSA). This includes a reduction in coagulase production, inhibiting their ability to grow on the traditional selective media *Staphylococcal* Medium 110, MSA, and Brilliance Blue [71,166]. Thus, the atypical morphologies and physiologies of SCVs present a challenge in the accurate clinical diagnosis of *S. aureus*. To avoid misidentification, extended conventional culturing and identification techniques must be implemented for the successful isolation of SCVs. Currently, culturing specimens on both blood agar and chromogenic *S. aureus* ID agar over 24–72 h incubation is the most rapid and accurate method of identifying *S. aureus* and their SCV phenotypes. Of note, SCVs have a growth rate of approximately one-ninth of typical *S. aureus*, causing them to be out-competed during the log growth phase [166]. To overcome this, isolates suspected of being *S. aureus* SCVs can undergo further confirmation through the use of molecular methods such as 16S rRNA partial sequencing [162], the amplification of species-specific DNA targets (coagulase, nuclease, *mecA* gene, *nuc* gene) [163], or an anti-Penicillin Binding Protein 2a (PBP2a) latex agglutination test [167]. However, the identification of SCVs is currently still a long process in comparison to the rapid diagnostics of typical *S. aureus* cell growth.

## 5. Future Directions and Diagnosis of *S. aureus* Lifestyles

*S. aureus*’s ability to switch to an alternative, more persistent cell type has been demonstrated to be an advantageous adaptation to alterations in the bacteria’s environment with persistent physical and chemical stresses. Previous studies have had a strong focus on different environmental stresses ranging from changes in pH, temperature, antibiotics, and a range of metabolic stresses. These changes within the host conditions cause the usually dominant, active *S. aureus* cells in a population to decrease in favour of a slower-growing, quasi-dormant phenotype, such as an SCV. The majority of these studies have focussed on characterising the changes in specific pathways as a result of mutations that alter the bacterial cell metabolism.

However, these studies are largely limited to *S. aureus* variations in cell types during infections, with very little analysis on these persistent SCV cell types or the proclivity for an active cell to switch to an SCV during the innocuous colonisation of healthy individuals. We postulate that some strains of *S. aureus* that asymptomatically colonise the different anatomical niches in the human host have the tendency to (1) grow as a single heterogenous population of cells that are made up of active cells, SCVs, and those predisposed to forming biofilms, and/or (2) have the ability to switch between the different alternative cell types during stress conditions.

In addition to this, the microbiomes of the various environmental niches within the human host exist as vast arrays of microorganisms that are constantly competing to establish themselves. The interactions between the different Staphylococci species have become a large focus in recent studies as part of the quest to identify molecules, such as bacteriocins, that can be utilised in place of or in conjunction with antimicrobial therapeutics as a potential strategy to combat the antibiotic resistance crisis.

## Figures and Tables

**Figure 1 antibiotics-13-00845-f001:**
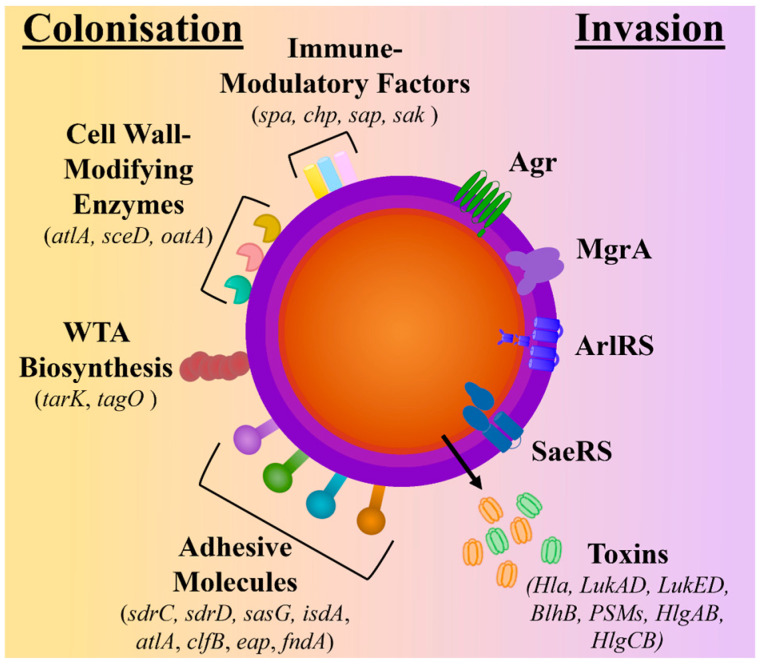
Comparison of molecular factors upregulated during colonisation and invasion. The schematic depicts the surface proteins known to be upregulated during colonisation and includes adhesive molecules, wall teichoic acid (WTA) biosynthesis molecules, cell wall-modifying enzymes, and immune-modulatory factors. In contrast, during invasion, the regulatory systems (Agr quorum-sensing systems ArlRS, SaeRS, and MgrA) are upregulated, which controls the expression and secretion of toxins.

**Figure 2 antibiotics-13-00845-f002:**
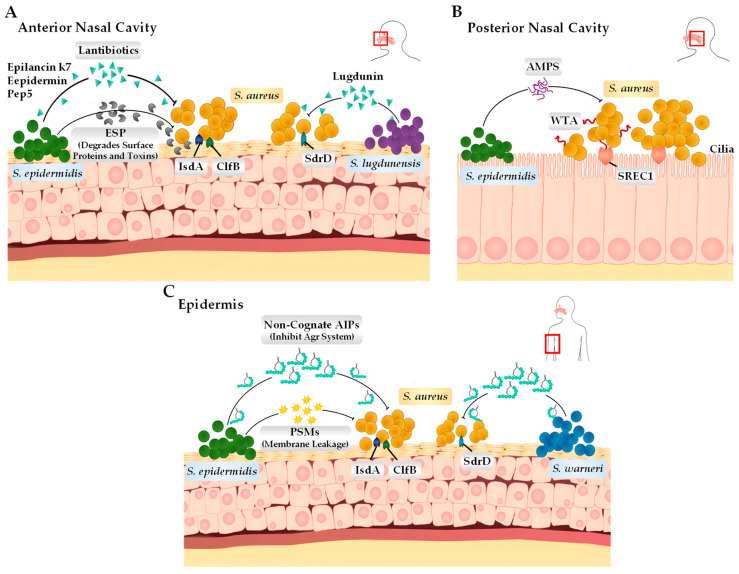
Bacterial adhesin and the interactions between coagulase-negative *Staphylococcus* species and *S. aureus* on the human epithelium. (**A**) In the anterior nasal cavity, *S. lugdunensis* and *S. epidermidis* produce lantibiotics (*S. lugdunensis* produces lugdunin and *S. epidermidis* produces epilancin k7, epidermin, and Pep5) that can directly kill *S. aureus*. Additionally, *S. epidermidis* can also produce extracellular serine protease that cleaves surface proteins such as IsdA, ClfB, and SdrD, interfering with the *S. aureus* ability to adhere to the epithelial surface. (**B**) In the posterior nasal cavity, *S. aureus* binds to the scavenger receptor class F membrane 1 (SREC1) through interactions with wall teichoic acid (WTA) present on the cell surface. *S. epidermidis* expresses its antimicrobial peptides (antibiotics and ESP) to either kill or inhibit *S. aureus* cells. (**C**) On the skin surface (epidermis), *S. warneri* and *S. epidermidis* produce autoinducing peptides (AIPs) through their Agr systems, which is deemed important for survival specifically on the skin surface. *S. epidermidis* can also produce phenol-soluble modulins (PSMs), which can disrupt the membrane of *S. aureus*, causing leakage.

**Figure 3 antibiotics-13-00845-f003:**
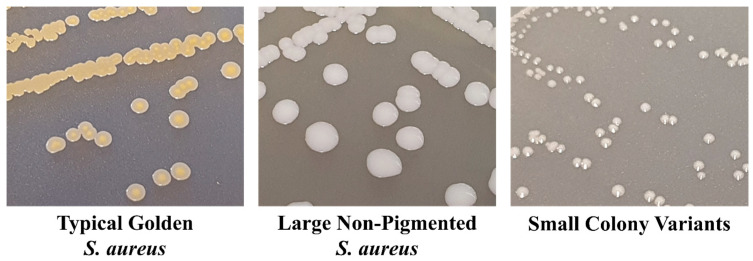
Comparison of typical *S. aureus* colonies, non-pigmented *S. aureus* colonies, and small colony variant (SCV) colonies grown on trypticase soya agar (TSA). Typical *S. aureus* produces large colonies with golden pigmentation, and in order to reduce the fitness cost, some strains of *S. aureus* present as large, non-pigmented colonies, whilst SCVs produce small, pinpoint colonies that are non-pigmented.

**Figure 4 antibiotics-13-00845-f004:**
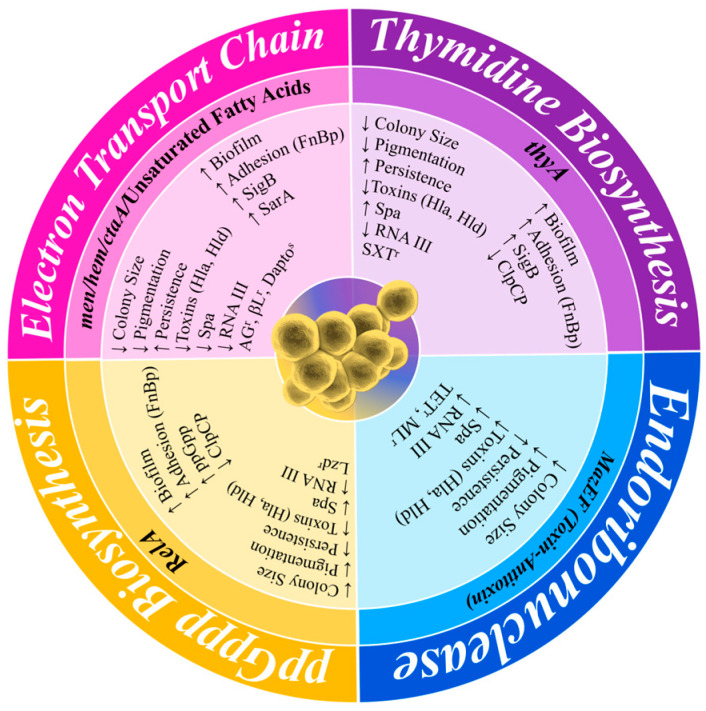
Comparison of the phenotype–genotype variation in SCVs resulting from different auxotrophic mutations. The outer layer of the figure shows the phenotypical change, the central layer shows the auxotrophic mutations, and the inner layer shows the phenotype of each mutation or specific gene and the increase and decrease in activity. Largely, this is displayed by the differences in their auxotrophic mutations. Hla, α-toxin; Hld, δ-toxin; Spa, Protein A; SigB, σ-factor B; SarA, Staphylococcal accessory regulator A; FnBP, fibronectin-binding protein; ClpCP, ATP-dependent protease C/P; AG^r^, aminoglycoside resistance; βL^r^, beta-lactam resistance; Dapto^s^, daptomycin susceptibility; TET^r^, tetracycline resistance; ML^r^, macrolide resistance.

**Figure 5 antibiotics-13-00845-f005:**
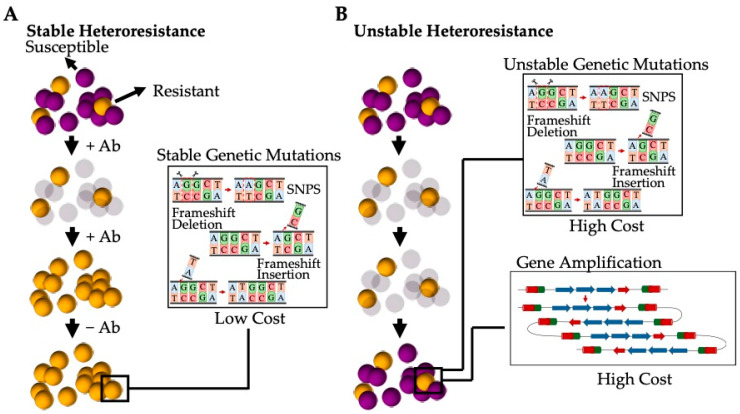
Heteroresistance generated by a subpopulation of bacterial cells present as a result of varying genetic mutations. (**A**) Stable heteroresistant mutations are a result of genetic mutations that have low or no fitness cost and that favour a stable phenotype that is resistant to varying antibiotics. These mutations are typically the result of SNPs, frameshifts, and insertions and/or deletions. (**B**) Unstable heteroresistant mutations are the result of genetic mutations that have a high fitness cost and that revert to the original antibiotic susceptible phenotype when the selective pressure of the antibiotic is removed. These mutations are typically a result of spontaneous, tandem amplification or high-cost SNPs, frameshifts, and insertions and/or deletions.

**Table 1 antibiotics-13-00845-t001:** Antimicrobial resistance genes and mechanisms in *Staphylococcus aureus*.

Class of Antibiotic	Genetic Basis	Mechanisms of Resistance
Cell Wall Synthesis Inhibitor		
β-lactams	*mecA*–SCCmec	Encodes PBP2a–reduces affinity for PBP
	*blaZ*–plasmid	Encodes β-lactamase–enzymatic hydrolysis of β-lactam nucleus
Glycopeptides	Gene unknown	VISA–alters peptidoglycan by thickening the cell wall, which traps vancomycin
	*vanRSHAXYZ*–from enterococci	VRSA–modifies the cell wall
Plasma Membrane Inhibitor		
Lipopeptide	*mprF*–gene mutation	Change in cell membrane charge–decreased drug binding
Protein Synthesis Inhibitor		
Aminoglycosides	*ant(4′)-I*–plasmid	Acetylating and/or phosphorylating enzymes that modify aminoglycosides
	*aph(3′)-III*–plasmid	
	*aac(6′)/aph(2”)*–plasmid	
Tetracyclines	*tetK*–plasmid *tetL*–plasmid	Active efflux
	*tetM*–plasmid *tetO*–plasmid	Ribosomal protection–competitive binding
Oxazolidinone	*cfr*–plasmid	Methylation of ribosome
	*rrn*	Mutations in 50s ribosomal L3 protein
Clindamycin, Macrolides, Streptogramins	*ermA*–plasmid	Encodes for ribosomal methylase that reduces binding to the 23s ribosomal subunit
	*ermB*–plasmid	
	*ermC*–plasmid	
	*msrA*	Active efflux
Nucleic Acid Inhibitor		
Fluoroquinolones	*parC/parE* (GrlA)–Topoisomerase IV	Mutations in the QRDR region, reducing the affinity of enzyme–DNA complex for quinolones
	*gyrA/gyrB*–DNA gyrase	
	*norA*	Active efflux
Rifampicin	*rpoB*	Reduces binding of rifampicin to RNA polymerase
Folic Acid Synthesis Inhibitor		
Trimethoprim/ Sulfamethoxazole	TMP–*dhfr* SMZ–*dhps*	Chromosomal mutations that cause target enzyme modification

TMP—trimethoprim; SMZ—sulfamethoxazole; VISA—vancomycin-intermediate *S. aureus*; VRSA—vancomycin-resistant *S. aureus*.

## Data Availability

Not applicable.

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
