# Peer review of "Phenotypic Variation in Staphylococcus aureus during Colonisation Involves Antibiotic-Tolerant Cell Types"

_antibiotics, 2024, doi:10.3390/antibiotics13090845_

Round 1

Reviewer 1 Report

Comments and Suggestions for Authors

This is an interesting review paper to introduce the phenotypic changes of S. aureus and its associated features including antibiotic tolerance. Although mostly well written, this reviewer would like to point out followings for improvement of this manuscript.

1. Figure 5 was shown in last part. However, authors mentioned SCV in earlier chapters. It is better to move it to earlier portion.

2. Figure 3. This figure is somewhat difficult to understand. Please improve explanation so that readers understand the meaning.

3. Figure legend of Figure 3. "Larely" is correct? "Spa, protein B" should be "Spa, protein A". 

4. In 2.2.1, 2.2., 2.2.3, some CoNS species were explained. However, S. capitis is missing. S. capitis is now important CoNS species in its increasing impact of infectious pathogen. S. capitis and some CoNS species can produce 6-TG that inhibit the growth of S. aureus. Authors should add a paragraph of S. capitis and 6^TG.

5. "hVISA" should be defined somewhere.

6. All the names of bacterial species should be italicized, throughout the manuscript.   

Comments on the Quality of English Language

Maybe because the authors are from Australia, the spelling of words is occasionally different from those used in standard English, e.g., "...sation". If it is necessary to adjust to a style of English, it is necessary to check whole manuscript.

Author Response

This is an interesting review paper to introduce the phenotypic changes of S. aureus and its associated features including antibiotic tolerance. Although mostly well written, this reviewer would like to point out followings for improvement of this manuscript.

  1. Figure 5 was shown in last part. However, authors mentioned SCV in earlier chapters. It is better to move it to earlier portion.

We thank the reviewer for this comment and now Figure 5 becomes Figure 3 and this has been added in the first introduction of SCV and the variations in the colonies– line 266 (fig 3 is then fig 4 and fig 4 is fig 5).

  1. Figure 3. This figure is somewhat difficult to understand. Please improve explanation so that readers understand the meaning.

We acknowledge this comment and we have added further description to the figure legend. It now explains the different layers (rings) of the figure.

  1. Figure legend of Figure 3. "Larely" is correct? "Spa, protein B" should be "Spa, protein A". 

This correction has been made.  

  1. In 2.2.1, 2.2., 2.2.3, some CoNS species were explained. However, S. capitis is missing. S. capitis is now important CoNS species in its increasing impact of infectious pathogen. S. capitis and some CoNS species can produce 6-TG that inhibit the growth of S. aureus. Authors should add a paragraph of S. capitis and 6^TG.

We thank the reviewer for this comment. While it is a larger field of study than can be fully described in this review, we had therefore focussed on specific CoNS. We do agree with this comment and have added another subsection (2.2.4) that provides the information on 6-TG.

  1. "hVISA" should be defined somewhere.

Text has been added to line 438 when hVISA is first mentioned.

  1. All the names of bacterial species should be italicized, throughout the manuscript.   

We have checked this: noting that in the headings/sub-heading that the normal text is italics the species names are non-italics.

Comments on the Quality of English Language

Maybe because the authors are from Australia, the spelling of words is occasionally different from those used in standard English, e.g., "...sation". If it is necessary to adjust to a style of English, it is necessary to check whole manuscript.

This has been checked and corrections made.

Reviewer 2 Report

Comments and Suggestions for Authors

Burford-Gorst & Kidd’s manuscript examines the phenotypic variation of Staphylococcus aureus during colonization, which is a relevant issue, given the clinical and epidemiological significance of S. aureus. However, the statement of the problem could be enhanced by providing more context on the clinical implications of these variations. For instance, discussing how these variations contribute to treatment challenges or chronic infections would strengthen the rationale for the study.

There are also several grammatical errors, some of which are listed below:

Line 14: “This dynamics in a S. aureus cell population…” should be rewritten as “These dynamics in S. aureus cell populations…” 

Line 299: A comma is needed before “and reduced fatty acid biosynthesis” for clarity.

Lines 320 to 321: The “fibronectin” should be preceded by “the” for consistency.

Lines 330 to 350: The “dependant” should be rewritten as “dependent”.

Line 457: The “make” in “The low population of these heteroresistant strains within a single isolate make it increasingly difficult to reliably determine an accurate susceptibility profile of the bacterial strain.” Should be rewritten as “makes”.

Line 557: The “However, for all these methods they do not take…” should be rewritten as “However, all these methods do not take…”, and a comma is needed before “which” to separate the non-restrictive clause.

Line 563: “deficiencies or reduction…” should be rewritten as “deficiencies or reductions…” for parallel structure.

Line 568: In the sentence “These atypical morphologies and physiologies of SCVs present a challenge in the accurate clinical diagnostics of S. aureus.”, “diagnostics” should be rewritten as “diagnosis”.

Comments on the Quality of English Language

Moderate edits are needed.

Author Response

Burford-Gorst & Kidd’s manuscript examines the phenotypic variation of Staphylococcus aureusduring colonization, which is a relevant issue, given the clinical and epidemiological significance of S. aureus. However, the statement of the problem could be enhanced by providing more context on the clinical implications of these variations. For instance, discussing how these variations contribute to treatment challenges or chronic infections would strengthen the rationale for the study.

Section 4.3 and 4.4 are a description of clinical application of the understanding of cell types and their impact on disease. We have added text (to 4.3) to emphasise the problem, and as the reviewer says, to make a clear statement of the problem and the clinical implications of these cell variations..

There are also several grammatical errors, some of which are listed below:

Line 14: “This dynamics in a S. aureus cell population…” should be rewritten as “These dynamics in S. aureus cell populations…” 

We thank the reviewer for this comment and have changed the text accordingly.

Line 299: A comma is needed before “and reduced fatty acid biosynthesis” for clarity.

We thank the reviewer for this comment and have changed the text accordingly.

Lines 320 to 321: The “fibronectin” should be preceded by “the” for consistency.

We thank the reviewer for this comment and have changed the text accordingly.

Lines 330 to 350: The “dependant” should be rewritten as “dependent”

We thank the reviewer for this comment and have changed the text accordingly.

Line 457: The “make” in “The low population of these heteroresistant strains within a single isolate make it increasingly difficult to reliably determine an accurate susceptibility profile of the bacterial strain.” Should be rewritten as “makes”.

We thank the reviewer for this comment and have changed the text accordingly.

Line 557: The “However, for all these methods they do not take…” should be rewritten as “However, all these methods do not take…”, and a comma is needed before “which” to separate the non-restrictive clause.

We thank the reviewer for this comment and have changed the text accordingly.

Line 563: “deficiencies or reduction…” should be rewritten as “deficiencies or reductions…” for parallel structure.

We thank the reviewer for this comment and have changed the text accordingly.

Line 568: In the sentence “These atypical morphologies and physiologies of SCVs present a challenge in the accurate clinical diagnostics of S. aureus.”, “diagnostics” should be rewritten as “diagnosis”

We thank the reviewer for this comment and have changed the text accordingly.